# Extrinsic Camera Calibration with Line-Laser Projection

**DOI:** 10.3390/s21041091

**Published:** 2021-02-05

**Authors:** Izaak Van Crombrugge, Rudi Penne, Steve Vanlanduit

**Affiliations:** Faculty of Applied Engineering Department Electromechanics, Universiteit Antwerpen, Groenenborgerlaan 171, 2020 Antwerpen, Belgium; Izaak.VanCrombrugge@uantwerpen.be (I.V.C.); Rudi.Penne@uantwerpen.be (R.P.)

**Keywords:** camera calibration, multi-camera, extrinsic calibration, non-overlap, field of view

## Abstract

Knowledge of precise camera poses is vital for multi-camera setups. Camera intrinsics can be obtained for each camera separately in lab conditions. For fixed multi-camera setups, the extrinsic calibration can only be done in situ. Usually, some markers are used, like checkerboards, requiring some level of overlap between cameras. In this work, we propose a method for cases with little or no overlap. Laser lines are projected on a plane (e.g., floor or wall) using a laser line projector. The pose of the plane and cameras is then optimized using bundle adjustment to match the lines seen by the cameras. To find the extrinsic calibration, only a partial overlap between the laser lines and the field of view of the cameras is needed. Real-world experiments were conducted both with and without overlapping fields of view, resulting in rotation errors below 0.5°. We show that the accuracy is comparable to other state-of-the-art methods while offering a more practical procedure. The method can also be used in large-scale applications and can be fully automated.

## 1. Introduction

Applications that use multi-camera setups require a good calibration between the cameras. Multiple cameras are used for a number of different reasons, e.g., to increase the observed area, for triangulation, and to resolve occlusions. For the latter two applications, overlapping fields of view (FoV) are required. For the first one, however, the overlap must be minimized or eliminated. An example of such an application is when objects or people are tracked across cameras [1,2]. So a good alignment of the images can be needed, even when the overlap is minimal.

Traditional calibration methods using markers like checkerboards [3] can be used directly when calibrating cameras with a large overlap. The problem becomes more challenging when the cameras share no overlap. In this work, we provide a solution to this problem: a scalable technique to determine the pose of a set of cameras with no need for overlapping views. It uses straight lines projected on a single planar surface. Each camera must see some part of this plane, but the parts do not need to overlap. The benefit of straight lines is that their correspondences across images can be used even when altogether different line segments are recorded in each image.

All calibration methods for non-overlapping cameras require some extra hardware. Our technique uses a line projector, which makes it highly suitable for calibration of surveillance cameras in large industrial scenes: with challenging lighting, many cameras, and minimal overlap for maximal FoV. In these scenes, the floor is the plane, as it is visible to all cameras. Industrial floors are generally quite flat, with deviations of no more than a few centimeters.

After an overview of the related work (Section 2), we present our algorithm (Section 3). To test the robustness under various conditions, we perform different sensitivity analyses using simulated images (Section 4). We also carry out two real-world experiments (Section 5). In the end, we compare our method to four other state-of-the-art methods (Section 6).

## 2. Related Work

When a multi-camera setup has overlapping views, the extrinsic calibration can be determined from a moving marker [4] or even from matching the moving pedestrians in view [5]. When there is no overlap, accurately determining the extrinsic calibration becomes more challenging. A wide variety of approaches to obtain the extrinsic calibration of cameras that share no overlap have been published [6].

A straightforward approach is to use a large physical target. They can be very simple, like a marked stick [7] or a sphere [8]. Or more convoluted calibration targets in the form of two checkerboards connected by a bar [9]. These objects have some disadvantages, despite their simple nature: it can be challenging to keep (part of) the calibration target in the view of all cameras simultaneously. The use of physical objects restricts the application to small and medium-scale camera setups as they do not scale well.

To connect the non-overlapping fields of view, an auxiliary camera can be added [10]. Xu et al. use mirrors to show the calibration target indirectly to multiple cameras [11]. An operator is needed to correctly aim the mirrors, which can be difficult in large scale applications. Using spherical mirrors makes aiming easier, but they reduce the image of the calibration target to a smaller resolution. This is bad for the calibration accuracy.

High-precision measurement instruments like laser trackers [12] or theodolites can be used for ultimate accuracy. However, they require specific know-how to operate and can be quite expensive.

A final category of techniques uses laser projection. The different fields of view are connected by a projection, like a laser dot or laser line. Previous methods used checkerboards [13,14,15]. Multiple checkerboard poses are needed for each camera, resulting in a laborious procedure that is hard to scale. The method we propose fits this last category, but it offers some key advantages. Compared to existing techniques, our approach does not require any manual intervention or expensive equipment. This makes it easily applicable in both small and large scenes.

## 3. Algorithm

The proposed method finds the extrinsic parameters of a set of *n* cameras. The intrinsic parameters of the cameras must be known. Intrinsic camera calibration is a well-known problem, and many good solutions already exist [3,16].

The calibration process is illustrated in Figure 1. There are five main steps:**Step** **A:**In the first step, laser lines are projected onto a plane and image sequences are captured by all cameras (Section 3.1).**Step** **B:**The calibration algorithm needs line correspondences between the different cameras. This can be a practical challenge, as it requires some form of synchronization between the cameras and the line projector. Our approach solves this problem by projecting the lines with different frequencies. Each line can then be isolated from the others by its specific frequency (Section 3.2) and is then detected in the image.**Step** **C:**A first estimate of the plane is made in optimization with only two degrees of freedom (Section 3.4). This results in a first estimate for the plane and the camera poses.**Step** **D:**Starting from this estimate, a refinement optimization is done (Section 3.5).**Step** **E:**The poses can only be determined up to a scale factor. Therefore, scaling may be needed (Section 3.6).

The cost function for the optimizations in Steps C and D is explained in Section 3.3.

### 3.1. Line Projection

The calibration algorithm is based on corresponding lines captured by the different cameras. Depending on the scene, the lines can be projected using different methods: line lasers, a rotating laser, a laser projector, a standard image projector, ... High power laser projectors are readily available as disco projectors, and they are designed to be easy to program. They use galvanometers to project any shape of the line and can be used at very large distances. For a fixed laser power, the intensity of the projected line decreases when the projection distance increases, due to the inverse-square law. However, the projected line does not need to be very bright. By modulating the laser lines (see Section 3.2), they can be reliably detected even if they are relatively dim.

The images must be captured in a way that allows the matching of the corresponding lines. There are different valid ways to accomplish this, of which we list three:**Manual:** A simple—but tedious—approach would be to project and capture one line at a time.This requires lots of manual intervention and may take quite some time. While this may be fine for a proof of concept, it may not be suited to calibrate many cameras in an active industrial environment where down-times must be kept to a minimum.**Triggered:** The cameras could be synchronized electronically with the projector. This means there must be a connection between the projector and the capturing system.**Modulated:** By modulating the different lines, the line projector can be a stand-alone device that can be moved around easily to different poses. Low modulation frequencies—in the range 1 to 3 Hz—can be used. This allows for simple projector hardware and works with all common camera frame rates. It works with all cameras—monochrome or RGB—that can record the projected wavelength.

We favor the approach with modulated lines because of its advantages, though the others are also valid. The choice of line capturing method, with equal carefulness, does not influence the calibration accuracy. However, it is a trade-off: more automation requires more technical complexity but speeds up capturing lines from many cameras. A more manual approach is easier to get started with, at the cost of a more laborious capturing process.

In our real-world experiments, we use two projection devices. We built a low-cost (under €25) laser projector with five standard 5 mW line lasers and an Arduino Micro, as shown in Figure 2. It can be powered by a small 9 V battery or a USB power bank. We also used an LED DLP projector (see Section 5.2) with a pre-generated video file of modulated lasers.

### 3.2. Line Correspondence by Frequency Separation

The intensity of each line is modulated in a square wave at a specific frequency per line. Other waveforms, like a sine, could also be used. A Fast Fourier Transform (FFT) will be used to separate the lines, so the detectable frequencies will be a multiple of $\Delta f=\frac{f_{s}}{N}$ with a maximum of $\frac{f_{s}}{2}$, where $f_{s}$ is the camera sample frequency and *N* is the total number of captured frames.

A perfect square wave only has odd-numbered harmonics. However, transient effects, non-linearities, and discretization effects in both the laser projector and the camera will result in even-numbered harmonics as well. Therefore the highest frequency is chosen to be smaller than double the lowest frequency to stay below the harmonics of the lowest frequency. We chose N=100 and $f_{s}=5$ Hz. The used frequencies are 1, 1.2, 1.4, 1.6, and 1.8 Hz.

Each camera records the images independently. By applying a per-pixel FFT (built-in optimized implementation of MATLAB [17]), an image of each individual line can be determined. This is illustrated in Figure 3.

Once the images are separated per line, each line has to be detected accurately from its image. This can be done using any robust line detection technique. We use a standard two-step method. First, a RANSAC 2D line estimation [18] is done on the 1% brightest pixels with 1000 iterations using a sample size of two points per iterations. The *N* inliers of this estimation are then used in the second step: a weighted least-squares refinement. This is a closed-loop solution that minimizes
(1)$$\sum_{i=1}^{N}v_{i}^{2}\cdot {\left|p_{i}l\right|}^{2}$$
with $v_{i}$ the intensity value of a pixel and $\left|p_{i}l\right|$ the perpendicular distance between this pixel and the line.

### 3.3. Bundle Adjustment Cost Function

The calibration process, illustrated in Figure 1, has two steps where bundle adjustment is used: the plane optimization in Step C (Section 3.4) and the pose refinement in Step D (Section 3.5). These optimizations use the same core cost function that takes a proposed plane and camera poses and returns a scalar cost value.

After Step B: Detect lines the 2D line projections in each camera image are known, as well as their endpoints. These are the endpoints of the line segment visible to the camera, so they may or may not lie on the image border. For a proposed plane and set of camera poses, the observations of one camera are projected back onto the assumed plane and then projected into the image of the other cameras. This reprojection is illustrated in Figure 4. If the assumed plane and poses approach the ground truth, then the reprojected lines will closely match the observed lines. The “distance” between two arbitrary 2D lines is ill-defined. Therefore, we choose to reproject the endpoints instead of the lines: the point-line distance has a clear geometrical meaning.

The involved cameras are assumed to be calibrated. This yields rectified image pixels $\left(u_{i},v_{i}\right)$ (that is, corrected for lens distortion), and known calibration matrices $K_{1},K_{2},\ldots$ for the individual cameras. A reference camera is chosen, named Camera 1. Each endpoint $p_{i}=\left(u_{i},v_{i}\right)$ in the Camera 1 image is reprojected onto the proposed plane π↔ax+by+cz+d=0 in point $P_{i}$. The world point $P_{i}$ is calculated: (2)$$\begin{array}{c} {\left(x_{i},y_{i},1\right)}^{⊤}=K_{1}^{-1}\cdot {\left(u_{i},v_{i},1\right)}^{⊤} \end{array}$$(3)$$\begin{array}{c} P_{i}=\left(X_{i},Y_{i},Z_{i}\right)=q\left(x_{i},y_{i},1\right)\;\mathrm{with}\;q=\frac{-d}{ax_{i}+by_{i}+c} \end{array}$$
$P_{i}$ is then projected onto the Camera *k* image: point $p_{i}^{\prime }=\left(u_{i}^{\prime },v_{i}^{\prime }\right)$. First, $P_{i}$ is transformed to Camera *k* coordinates via $T_{k}=\left(R_{k}|t_{k}\right)$ and then projected using the intrinsic matrix $K_{k}$ of Camera *k*: (4)$$w_{i}{\left(u_{i}^{\prime },v_{i}^{\prime },1\right)}^{⊤}=K_{k}T_{k}{\left(X_{i},Y_{i},Z_{i}\right)}^{⊤}$$

The cost for this endpoint is ${\left|p_{i}^{\prime }l_{i}^{\prime }\right|}^{2}$, the squared Euclidean distance between $p_{i}^{\prime }$ and the corresponding line $l_{i}^{\prime }$ in the Camera *k* image. The total cost for a given plane and set of camera poses is the sum over all cameras $\sum {\left|p_{i}^{\prime }l_{i}^{\prime }\right|}^{2}$ for all endpoints of Camera 1 reprojected to each Camera *k*.

### 3.4. Plane Optimization

The axes of Camera 1 are used as the reference coordinate system. A plane has three degrees of freedom (3 DoF). For the optimizations, the plane is represented using the following independent parameters:α: the angle between the normal of the plane and the yz-plane.β: the angle between the normal of the plane and the xz-plane.*d*: the distance from the plane to the origin.

Because monocular cameras are used, the scale can not be found in this optimization. Therefore *d* is set to 1. The optimizer varies only α and β. The three-plane parameters of the form π↔ax+by+cz+d=0 can be found as such: (5)$$\left(a,b,c\right)=\frac{n}{∥n∥}\;\mathrm{with}\;n=\left(\tan\alpha ,\tan\beta ,1\right)$$

In Step C: Optimize plane (Figure 1) the plane is estimated using an optimization with only two degrees of freedom (α and β). Everything will be scaled to the correct size in Step E: Scale poses.

For each tested set of plane parameters, all transformations are estimated from the homography between the plane lines $L_{i}$ and the image lines $l_{i}^{\prime }$. This homography can be found using a Direct Linear Transform [19] based on the homogeneous line coordinates: $\left(F_{i},G_{i},1\right)$ for $L_{i}$ and $\left(f_{i}^{\prime },g_{i}^{\prime },1\right)$ for $l_{i}^{\prime }$. Once we find the 2D representation of the endpoints $P_{i}$ in the plane π, we can easily find $\left(F_{i},G_{i},1\right)$.

One way to implement this is the following. First, a rigid transformation $T_{xy}$ is determined that makes the plane π parallel to the xy-plane in Camera 1 coordinates. When $P_{i}$ are transformed by $T_{xy}$, their *z*-coordinate will be constant. From the endpoints $P_{i}$, the homogeneous line parameters $\left(F_{i},G_{i},1\right)$ in the plane are found in the form $L_{i}↔F_{i}x+G_{i}y+1=0$. Because the line parameters $\left(f_{i}^{\prime },g_{i}^{\prime },1\right)$ of the projected line $l_{i}^{\prime }$ are already known, the homography can be determined. The camera poses can be calculated directly from this homography. The same technique to determine the camera poses from the homography as in [20] is used, given that the homography $p_{i}^{\prime }=HP_{i}$ between points corresponds to $l_{i}^{\prime }=H^{⊤}L_{i}$ for lines. The camera poses relative to Camera 1 are found by transforming the resulting poses by $T_{xy}^{-1}$.

The reprojection error $\sum {\left|p_{i}^{\prime }l_{i}^{\prime }\right|}^{2}$ (calculated as described in Section 3.3) is minimized by varying α and β. By optimizing only these 2 DoF, a plane is found that is optimal for all cameras. The plane optimization also yields a first estimate for the camera poses.

The cost function $\sum {\left|p_{i}^{\prime }l_{i}^{\prime }\right|}^{2}$ typically has multiple local minima, while there is only one true solution. To find the global minimum, this optimization step is done using the MultiStart algorithm from the MATLAB Global Optimization Toolbox. The local optimizations are done as constrained optimizations with the Interior Point Algorithm using fmincon. The initial values for α and β are $-\frac{\pi }{4}$, 0, and $\frac{\pi }{4}$. So 9 start points are used: $\left(-\frac{\pi }{4},-\frac{\pi }{4}\right);\left(-\frac{\pi }{4},0\right);\cdots ;\left(\frac{\pi }{4},0\right);\left(\frac{\pi }{4},\frac{\pi }{4}\right)$. Our experiments show that the use of MultiStart is sufficient to find the global optimum. Although the local optimization is run 9 times now, the plane optimization is still fast because there are only 2 DoF.

### 3.5. Pose Refinement

Each camera pose is refined in Step D: Refine poses (Figure 1). For *n* cameras, this optimization has 2+6(n−1) degrees of freedom: two for the plane and six for each of the n−1 cameras, as Camera 1 is the reference. Its pose is the identity transformation $I_{4}$. Despite the higher number of DoF, this optimization converges quickly thanks to the good initial estimate. The resulting plane and poses are: (6)$$\pi ,T_{2},T_{3},\ldots ,T_{k}=\underset{\pi ,T_{2},T_{3},\ldots ,T_{k}}{\mathrm{argmin}}\sum {\left|p_{i}^{\prime }l_{i}^{\prime }\right|}^{2}.$$

### 3.6. Pose Scaling

After the pose refinement, all relative camera poses are known. For many applications this is sufficient. Nevertheless, sometimes the poses are needed in world units. A final scaling step can be added for those applications that need an absolute scale. We list a few examples of the different ways to determine the scale:When the distance is known between one of the camera centers and the projection plane. If the projection plane is the floor, then this distance is the height of that camera. This is used in all simulation experiments of Section 4. Usually, the exact location of the camera center is known with an uncertainty of several millimeters. However, the distance between the camera center and the plane is several orders of magnitude larger. So the location error of the exact camera center will not contribute significantly to the total scale error.If the real-world distance between two cameras is known, the scale is known. This is used in the experiment with the translation stage in Section 5.1.If one or more stereo cameras are involved, its stereo baseline determines the scale.A marker of known size can be applied to the projection plane or two markers with a known distance between them. Such markers can be detected either automatically or manually in a camera image. Knowing the parameters of the calculated plane, the image points can be reprojected to calculate the scale. When using the floor plane, this could also be the known size between the seams of the floor tiles or an object of known size placed on the floor.

## 4. Evaluation on Simulated Data

We conduct a number of experiments on simulated data, as well as on real-world images (Section 5). All experiments are evaluated using the same metrics. The simulations are used for a series of sensitivity analyses.

### 4.1. Evaluation Metrics

It is common to use the reprojection errors as evaluation metrics. While it is the correct measure to assess the correspondence between the measurements and the reconstructed model, it is not always indicative of the correspondence between this model and the ground truth [21]. Furthermore, sometimes the reprojection error is inversely proportional to the actual accuracy. This can be seen in our penultimate experiment: in Figure 12 it can be seen that the reprojection error after refinement increases with the number of lines while the actual rotation error decreases. We still report the reprojection error in this manuscript for comparison and because it shows how good the optimization converged.

In our opinion, the rotation error is the most meaningful metric for accuracy. This is the difference in rotation between the estimated camera pose and the ground truth. The rotation error leads to translation errors proportional to the camera distance. Given the ground truth rotation matrix $R_{i,GT}$ and the calculated camera rotation matrix $R_{i}$ of camera pose *i*, the rotation error is the angle in the axis-angle representation of $R_{i}R_{i,GT}^{-1}$.

The tolerance for rotation error is application dependent. In our experience, a ‘good’ result is when the rotation error is below 0.5°. This is similar to state-of-the-art methods as described in Section 6. In most experiments, the results are reported before and after refinement. These are the transformations before and after Step D: Refine poses of Figure 1.

### 4.2. Sensitivity Analysis with Simulation

Several are performed on simulated data to evaluate the robustness of the algorithm to different error sources. In each experiment, a different error is introduced without changing the other parameters. In all simulations, a scene is used with no FoV overlap (Figure 5b), and intensity noise is added to simulate sensor noise unless otherwise specified. The noise is Gaussian with σ=0.01, given that the pixel values lie between 0 and 1. The used scenes are described in Section 4.3. The virtual cameras are modeled after the left camera of the Intel RealSense D415: a horizontal viewing angle of 65° and a resolution of 1920 × 1080.

### 4.3. Comparison between Overlapping and Non-Overlapping Fields of View

The purpose of the proposed method is to provide a good extrinsic calibration, even when there is no overlap in the field of view of the cameras. To validate this, two scenes are made: one with and one without camera overlap. The scenes are made in Blender 2.82: an open source 3D modelling and rendering package, available at www.blender.org. In the first scene (Figure 5a) seven cameras are pointed at the same point on a wall, sharing a considerable overlap in FoV. The distance between the cameras and the wall varies between 1 m and 5 m. In the second scene (Figure 5b) six cameras are pointed at the wall, all at a distance of 3 m. Cameras 3 and 4 look at the wall perpendicularly. The other cameras in this scene are angled at 15°.

The experiment was run with six lines for the first scene and 12 lines for the second scene. The second scene needs more lines so that enough lines are shared between the cameras. For comparability, the lines were chosen so that, on average, each camera also sees six lines.

The results in Figure 6 confirm that no overlap is needed to obtain a good extrinsic calibration. After the refinement, the rotation errors are all below 0.15°. The final results with and without overlap are very similar. The large errors in the rough estimate for the scene with no overlap demonstrate the need for a refinement step when there is no overlap. When there is plenty of overlap, the refinement step no longer has a significant contribution.

### 4.4. Sensitivity to Sensor Noise

All cameras have some sensor noise. To verify the robustness of the method against this noise, Gaussian noise is added to the images before line detection. The standard deviation σ of the noise is varied between 0 and 0.1, given that the pixel values range from 0 to 1.

It can be seen in Figure 7 that there is no correlation between the image noise level and the resulting accuracy. This is as expected: the line detection is done on the entire line. Because many image points are used, the noise is dealt with robustly.

### 4.5. Sensitivity to Errors in Camera Intrinsics

While camera intrinsics can be determined quite accurately, they are not always perfect. A robust extrinsic calibration method should not be too sensitive to errors in the intrinsics. Two intrinsic parameters are evaluated: errors in the focal length and errors in the principal point. The focal length and the principal point of the assumed camera model are varied. Only the effect of the horizontal coordinate of the principal point is shown here because the effect of the vertical coordinate is analogous.

Instead of the correct focal length $f_{GT}$, a different value *f* is used. The same is done for $c_{x}$, the horizontal component of the principal point. Given that *W* is the width of the image, the relative focal length error $\epsilon_{f}$ and relative principal point error $\epsilon_{pp}$ are expressed as:(7)$$\epsilon_{f}=\frac{f-f_{GT}}{f_{GT}}\cdot 100\%\;\mathrm{and}\;\epsilon_{pp}=\frac{c_{x}-c_{x,GT}}{W}\cdot 100\%$$

Our experiments show a clear correlation between the rotation error and the translation error. For conciseness, we only report the rotation error for this experiment, as this error metric is the most indicative of the accuracy in applications.

As expected, the rotation errors increase with increasing (absolute) focal length and principal point errors. Figure 8 shows that there is more sensitivity to principal point errors than to focal length errors. To keep the median rotation error below 0.5°, the absolute focal length error must not exceed 2%, whereas the absolute principal point error must not exceed 1%.

### 4.6. Sensitivity to Plane Curvature

The optimization relies on the assumption of a flat surface. A real-world surface will have some (small) amount of curvature, which results in errors. The same experiment as before is done where the plane has different amounts of curvature.

The simulated surface is a square, as shown in Figure 5b, and its sides measure 8 m. A cylindrical curvature (Figure 9a) is introduced with its axis parallel to the *x*-axis. In Figure 9b,c, the curvature is reported as an angle (α) in degrees. The curvature radius is therefore $R=\frac{8\;m\cdot 180^{\circ }}{\alpha \cdot \pi }$. It must be noted that the curvature radius should be interpreted in the context of the scene scale.

Figure 9b,c show that the method is sensitive to plane curvature when the cameras share no overlapping FoV. To keep the median rotation error below 0.5°, the plane curvature should not exceed 1°. This corresponds to a curvature radius of over 450 m.

However, the method is very robust to plane curvature when the cameras share significant overlap. The rotation error stays below 0.4° up to a plane curvature of 5°, corresponding with a curvature radius below 100 m. It must be noted that in urban or industrial scenes, there are many surfaces with little curvature.

### 4.7. Sensitivity to Line Curvature

Just like the plane, the lines are assumed to be straight. When the projected lines are not perfectly straight, the accuracy will decrease. For this sensitivity analysis, the same experiment is done with different line curvatures. The line curvature is defined analogous to the plane curvature, as shown in Figure 9a.

Figure 10 shows a strong sensitivity to line curvature. To keep the median rotation error below 0.5°, the line curvature should not exceed 0.5° for without overlap and 2° with overlap. The straightness requirement for the lines is stricter than that for the plane. Fortunately, optical instruments like lasers or other projectors can easily produce lines with no significant curvature.

## 5. Real-World Experiments

For the real-world experiments, the same evaluation metrics are used as those used for the simulations, as set out in Section 4.1. Two different experiments are performed. In the first one, there is some overlap in the fields of view (Section 5.1). In the second one, there is no overlap at all (Section 5.2).

Intel RealSense D415 cameras are used. They have a horizontal viewing angle of 65° and the images are captured with a resolution of 1920 × 1080. They are stereo cameras, but only the left camera is used.

### 5.1. Translation Stage for Ground Truth

To obtain exact poses, a single camera is mounted on a high-precision translation stage. The camera is aimed perpendicular to the translation motion. A Zaber X-LRT1500AL-C-KX14C translation stage is used, with an accuracy of 375 µm. The camera is translated to 5 different poses at equal distances of 300 mm. The translation stage movement provides the ground truth: exact translation and no rotation (Figure 11).

Five laser lines are projected using the self-made laser projector shown in Figure 2. The measurements are done twice with different laser line positions to obtain 10 different lines. To test the influence of the number of lines used, the calibration is done multiple times, each time using a different number of lines. The different camera poses have a significant amount of FoV overlap.

The rotation error goes down as the number of lines increases (Figure 12). When using eight or more lines, the rotation errors after refinement are—for the large majority—below 0.5°.

### 5.2. 360° Camera Setup

To demonstrate a real-world application with no FoV overlap, a 360° camera setup is built, as shown in Figure 13a. Four Intel RealSense D415 cameras are mounted on a square looking out. They have a horizontal view angle of 65°, so they do not overlap. Using a small LED DLP projector (Optoma ML750), 9 lines are projected on a light-colored wall (Figure 13b). The setup is positioned at a distance of about 0.6 m from the wall. Two cameras at a time are aimed so that they both look at the wall at roughly 45°. After recording the lines with this camera pair, the setup is rotated by about 90 degrees to aim the next pair of cameras. In this manner, four recordings are made, one for each combination of adjacent cameras.

The transformation between each couple of cameras is calculated with the proposed method. The product of the four pairwise transformations should be the identity transformation $I_{4}$ as the loop is closed. The rotation and translation errors are calculated from this closed loop. The resulting RMS (Root Mean Square) reprojection error is 0.22 pixels. The combined rotation error of the four transformations is 1.86° and the translation error is 33.3 mm.

## 6. Comparison to State of the Art

The state of the art in camera calibration without overlapping views is very diverse. As a result, there is no standardized procedure that allows a truly fair comparison. We refrain from using the reprojection errors, as explained in Section 4.1. Instead, the rotation errors are shown in Table 1.

We selected methods with the same goal: obtaining the extrinsic calibration parameters for cameras with no overlap. The compared experiments all use scenes of comparable scale, where the distances between the cameras and the calibration targets are between 1 and 10 m. They all use the same metric: rotation error. To make the comparison as fair as possible, the results from the original publications are used. Following methods are included in the comparison:In the work of Liu et al. [14], laser planes are visualized in different cameras by using line lasers and checkerboards. They use simulations as well as real-world measurements. The rotation error for their real-world experiments shown in Table 1 is calculated from the Euler angles they reported.Van Crombrugge et al. [20] use a standard LCD or DLP projector to project Gray code. We report the median rotation errors for simulation with no overlap and three real-world experiments, hence the range instead of a single number.Robinson et al. [22] use a straightforward method. The two non-overlapping cameras each have a checkerboard in view. A third camera is temporarily added that has both checkerboards in view. No results for real-world experiments were published, only simulations.Zhu et al. [23] use “planar structures in the scene and combine plane-based structure from motion, camera pose estimation, and task-specific bundle adjustment.” The rotation error reported here is the mean pose error of 16 cameras compared to the ground truth.

For the proposed method, we report the median rotation error after refinement. The simulation results are those of the scene with no overlap. The real-world results are of the translation stage experiment in Section 5.1 when using 8 lines.

The comparison in Table 1 shows that the performance of the proposed method is similar to that of other state-of-the-art methods. It does not offer superior accuracy but does provide a straightforward and practical calibration process that is scalable. When comparing the ease of use of the different methods, the proposed method shows some clear advantages. The only extra hardware needed is a device that can project straight laser lines. The calibration process can be done entirely automatically.

In comparison, the technique of Robinson et al. [22] requires an extra camera and–more importantly—many checkerboards or a single checkerboard that must be moved manually. This hinders scalability, as more and larger checkerboards would be needed to increase the number of cameras and the scene scale, respectively.

The technique of Liu et al. [14] has a more labor-intensive calibration procedure. To obtain the light plane, a minimum of two checkerboard poses is needed. A checkerboard must be held in the view of each camera at least six times to obtain the required three (or more) light planes. This becomes cumbersome when a larger number of cameras is used.

Because the method of Zhu et al. [23] uses structure from motion, it can only be used for camera setups that can be moved in one piece. The result reported here was the result of varying the pose of the multi-camera setup 40 times. This asks much manual effort of the operator, and the procedure can not be used for immovable camera setups.

## 7. Conclusions

In this article, we proposed and validated a novel technique to determine the extrinsic parameters of a set of cameras. The technique is applicable even if the cameras share no overlap in their fields of view. It can only be applied in cases where the cameras see a shared planar surface. As a minimum, each camera should at least see four different lines. For good results, we suggest using at least eight lines.

Sensitivity analysis showed good robustness against image noise. It also showed that accurate intrinsic calibration is needed to get good extrinsic calibration results. There is some tolerance for plane curvature, especially if the cameras share a large overlap. When there is no overlap, the plane curvature should not exceed 1°. The straightness of the lines is more critical, but this should not be an issue for most projectors. The accuracy was confirmed in real-world experiments, both with and without overlap.

The main advantages of this technique compared to other calibration methods for non-overlapping cameras are that it has a practical automated procedure and that it is scalable. A large number of cameras can be calibrated efficiently because no manual intervention is needed. By using projection and frequency modulation, it can also be used in large scale scenes with challenging ambient light. This makes the method especially suitable for surveillance networks. The accuracy is less than with most extrinsic calibration methods. However, compared to methods that also work with no overlap, it is similar to the state of the art. The only prerequisites are a line projector and a good intrinsic calibration of the cameras.

## Figures and Tables

**Figure 1 sensors-21-01091-f001:**
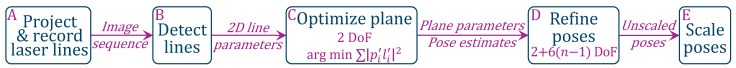
Block diagram of the proposed calibration method.

**Figure 2 sensors-21-01091-f002:**
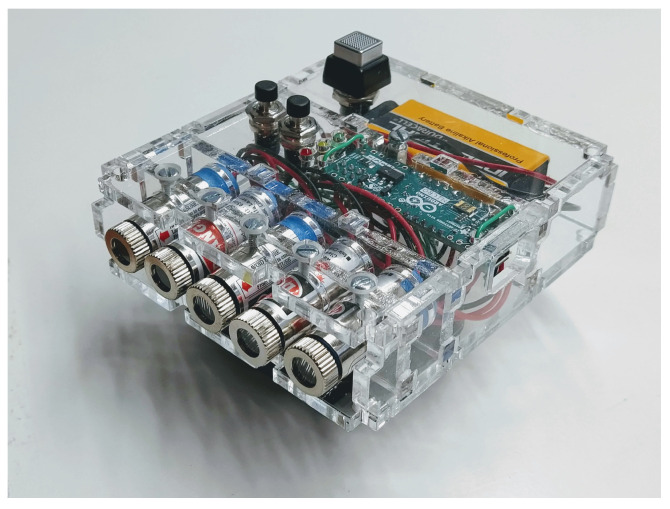
Example of a low-cost laser projector.

**Figure 3 sensors-21-01091-f003:**
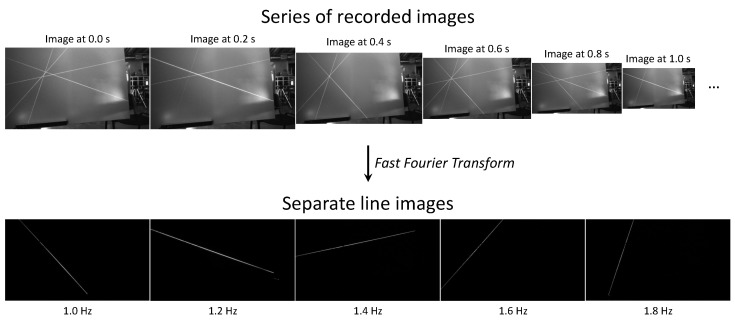
Different lines are projected at different frequencies in a square wave. 100 frames are recorded with a sampling period of 0.2 s. A per-pixel Fast Fourier Transform (FFT) is used to separate the lines.

**Figure 4 sensors-21-01091-f004:**
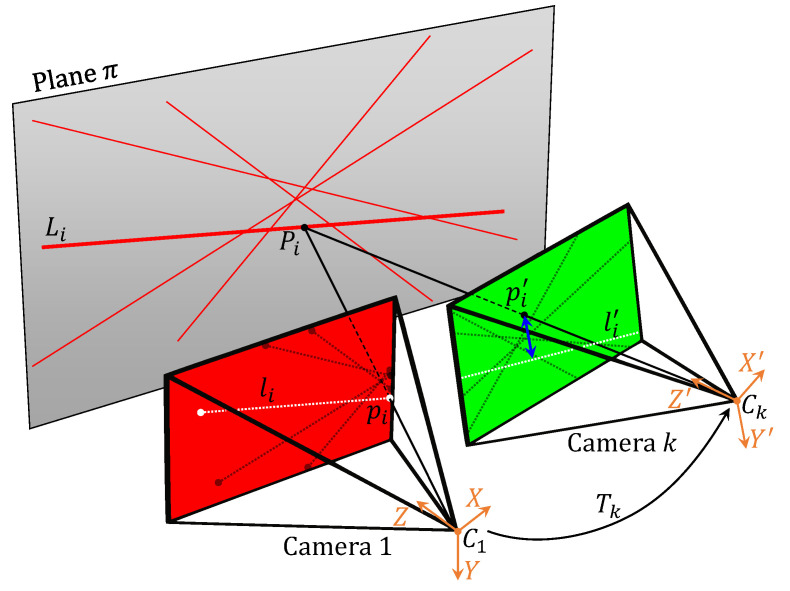
Illustration of the reprojection of an endpoint from Camera 1 to Camera *k* for a given plane and relative pose $T_{k}$.

**Figure 5 sensors-21-01091-f005:**
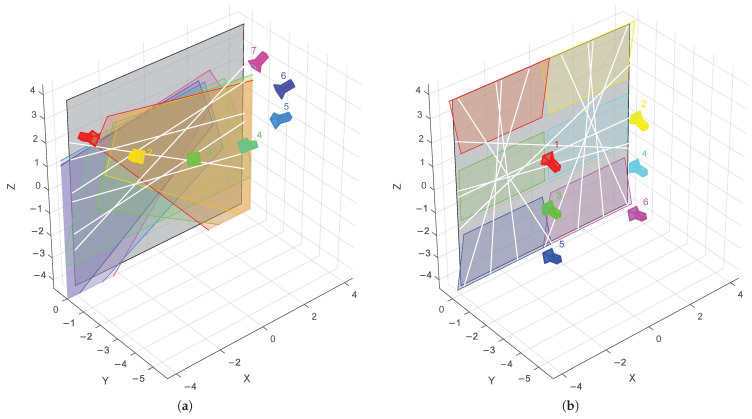
The two simulation scenes: (**a**) Scene with 7 overlapping cameras. (**b**) Scene with 6 non-overlapping cameras. The wall is shown in gray, the fields of view (FoV) of each camera is shown in its corresponding color, and the laser lines are shown in white.

**Figure 6 sensors-21-01091-f006:**
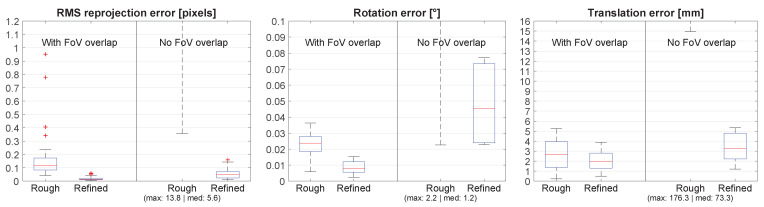
Comparison between scenes with and without overlap. Rough and Refined refer to the results before and after the refinement step. Missing maximum and median values are shown in parentheses.

**Figure 7 sensors-21-01091-f007:**
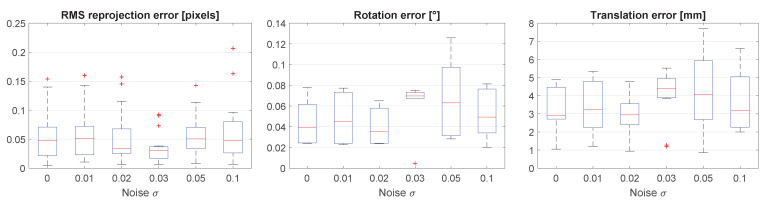
Results when varying the standard deviation σ of the Gaussian noise.

**Figure 8 sensors-21-01091-f008:**
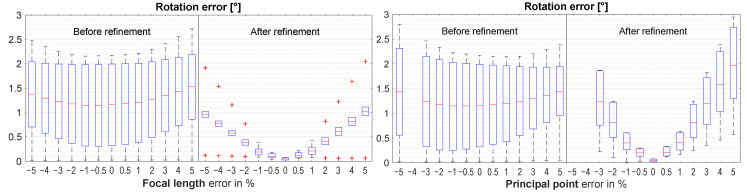
Results when introducing errors in camera intrinsics. Box plots are missing where the optimization did not converge.

**Figure 9 sensors-21-01091-f009:**
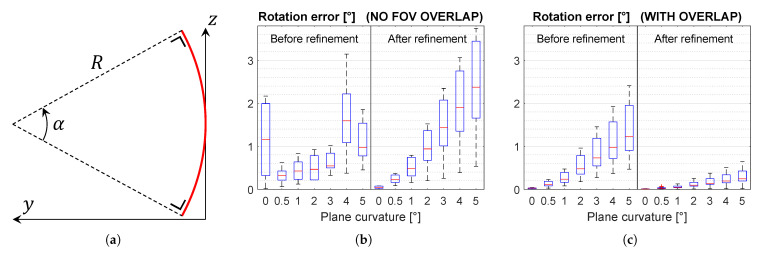
(**a**) Schematic illustration of the plane curvature. The red arc is the cylindrically curved plane in side view. (**b**) Results when varying the plane curvature for the scene with no overlap and (**c**) for the scene with overlap.

**Figure 10 sensors-21-01091-f010:**
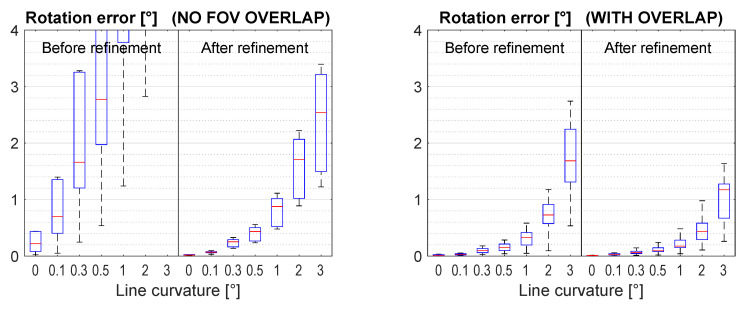
Results when varying the plane curvature for the scene without and with overlap.

**Figure 11 sensors-21-01091-f011:**
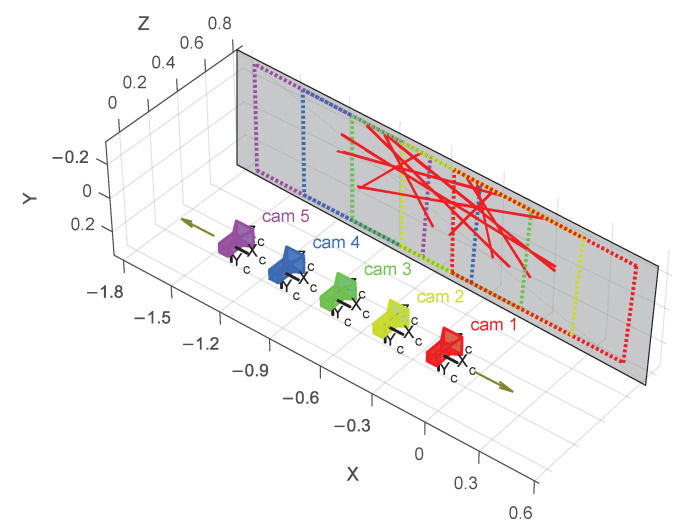
The five poses of the camera, the position of the plane, and the 10 projected lines. The respective fields of view are shown in a dashed line. The arrows show the direction of travel of the translation stage.

**Figure 12 sensors-21-01091-f012:**
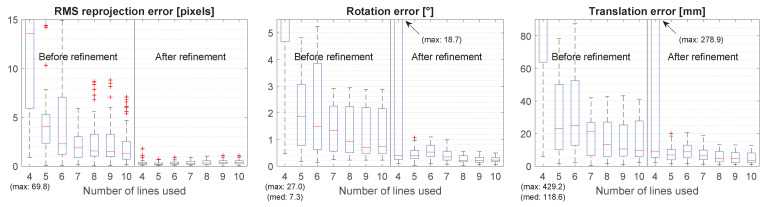
Box plots of the results with the translation stage as ground truth. The first row shows the results before refinement and the second row shows the results after refinement. The calibration errors with four lines are not fully visible because they are so much larger than the rest. Missing maximum and median values are shown in parentheses.

**Figure 13 sensors-21-01091-f013:**
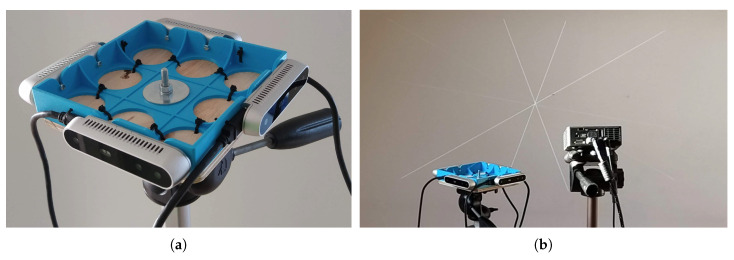
(**a**) The 360° camera setup with four Intel RealSense D415 cameras. (**b**) The complete measurement setup with cameras, projector, and projected lines.

**Table 1 sensors-21-01091-t001:** Comparison of five state-of-the-art methods. Because the validation experiments vary greatly between the different techniques, the results can not be compared directly. They do, however, indicate the order of magnitude of the obtained accuracy.

Using Simulated Images		Using Real-World Images	
**Method**	**Rotation Error [°]**	**Method**	**Rotation Error [°]**
Liu et al. [14]	<0.005	Liu et al. [14]	0.016
Van Crombrugge et al. [20]	< 0.015	Van Crombrugge et al. [20]	0.121–0.362
Robinson et al. [22]	< 0.04	Zhu et al. [23]	0.688
Proposed method	< 0.045	Proposed method	0.22

## Data Availability

Not applicable.

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
