# Peer review of "Extrinsic Camera Calibration with Line-Laser Projection"

_sensors, 2021, doi:10.3390/s21041091_

Round 1

Reviewer 1 Report

This paper proposes algorithm and procedure for calibration (extrinsic mutual calibration) of many cameras with limited size of common field of view. The proposed approach is mixture of existing solutions and from scientific point of view presents limited novelty. However, the technological and experimental work is very described with high details, well explained and compared with existing approaches and provides high practical value for community.

I reviewed the previous version of the article, hence I do not have many detailed comments.

Detailed comments:

Line 9 – “rotation errors below 0.5.” – please specify unit of measure.

Author Response

Thank you for your review.

We have made some changes to the source of the manuscript to resolve the problem of the missing character tokens.

Reviewer 2 Report

Remarks on the research:

  • The expectancy of the work in term of accuracy is not very high, excluding photogrammetric applications and aiming more toward simple imaging applications of the "streetview" type. It would be interesting in a future work to analyse more precisely the sources of errors and find solutions to overcome them to improve, if possible, the accuracy of this method to a photogrammetric level.
  • Finding a wall big and flat enough might be difficult.
  • If the imaging system is stable enough, the method should work with successive single lines, that would ease the detection process, by making the FFT process that I suppose is very ressource consuming unnecessary. Moreover, the device projecting the line could be a lot simpler.
  • The straightness of the lines is assumed but not tested, nor the effect of the not straightness by simulation.
  • Line 224 you state : "the rotation error is the most meaningful metric for accuracy" but evaluation of accuracy by only the angle doesn't evaluate the accuracy of the scaling phase witch can be an important source of error.

Remarks on the paper:

The paper is clear and well written, but some problems remains :

Lines 9, 100, 230, 240, 248, 253, 275, 286, 289, 296, 297, 311, 315, 318, 324, 377 and  in Table 1 and maybe other: Some symbols seem to be missing in the downloaded paper such as ° or €, leading to values without unit for all angles. In the same way, the resolution of the cameras appears as 19201080 in place of 1920x1080.

The measure of the plane curvature by an angle is not very clear in my opinion; could you use another way to quantify it, or else give a geometric meaning to this angle ?

Author Response

This manuscript is a resubmission of an earlier submission. The following is a list of the peer review reports and author responses from that submission.

Round 1

Reviewer 1 Report

This paper proposes algorithm and procedure for calibration (extrinsic mutual calibration) of many cameras. The proposed approach is mixture of existing solutions. I see very limited novelty, mainly practical and experimental one. I think that this paper is not suitable for Sensors journal.

Detailed comments:

Lines 51-53 – this sentence is unclear. You could calibrate set of static cameras using SfM technique.

Lines 54-59 – Please elaborate situations when all calibrated cameras are directed to one surface with projections versus situations when some cameras are calibrated on floor but other set of cameras are directed to ceiling?

Fig. 1 – this figure can be removed. It is repetition of steps defined from line 67

Line 74 – “pane” -> plane

Line 101 – “least square refinement” – please elaborate on algorithm. There is many algorithms for line fitting.

Line 106 – “After Step B the 2D lines in each camera image are known” – lines or lines projections? Endpoints on image border or real endpoints of lines? There is uncertainty between real line endpoints or endpoints projection or endpoint on image border, please use precise description.

Section 3.2 – this criterion is very subjective and seems to be not the best one. Why not use reprojections of whole lines? In my opinion it is weak point of this work.

Line 140 – what exactly means “pose I4.”

Section 3.5: How do you know nodal point (virtual point) distance to floor? What exactly means distance between two cameras?

There is many typos, please carefully correct.

Reviewer 2 Report

This is a good paper, introducing a novel approach for automatic extrinsic calibration of cameras based on laser lines projected on a plane. It is well-structured, well-documented and includes adequate experimental results, both simulated and real-world. Their approach for easily establishing line correspondences is worth noting.

My comments are as follows.

1. A main aspect of this work is its suitability for extrinsic calibration of cameras with non-overlapping fields of view. In fact, it exploits a fundamental advantage of straight linear features whose correspondences across images may be established (and metrically used) even when altogether different line segments are recorded in each image. This should be explicitly stressed in the text.

2. In Section 4.2 the authors use line intersections for calculating the camera poses (translation). The reader may think that these intersections must be actually visible on all images (thus challenging the assumption of non-overlapping cameras). This in not the case since the image lines can be extended and intersected even outside the actual image format. This clarification would be welcome.

3. Regarding scaling, I find the last suggestion in Section 3.5 as the most realistic. These markers could be suitable for automatic extraction, thus avoiding user interference.

4. I suggest to show some images from the real-world experiments.

5. Also:

Line 39: “Or more convoluted calibration targets in the form of two checkerboards connected by a bar [9]”. I think that this paper does not use checkerboards!

Line 223: Write “Figure 6(b)” instead of “Figure 6(a)”.

Reviewer 3 Report

The paper proposes an extrinsic calibration procedure with line projection laser allowing to calibrate the external parameters of multiple cameras at once, even without any superposition of the views.

The topic of camera calibration has already lots of work but given its relevance the paper is interesting in particular because it proposes a method that can solve the calibration of several cameras that are not viewing the same area (this could be interesting in several application such as surveillance for example).

The paper is interesting and scientifically sound. The main question I have are related to the mono vs stereo methods.  The calibration of stereo camera is an extension of monocular calibration (in the limit one could calibrate each camera of the pair and then refine the initial results). The main novelty of the paper is on the use of the projected line and thus I think the paper would be simpler and with the same impact if presenting only the monocular calibration.

The other limitation of the approach is related to the necessity to have specific HW (a mounting able to project many lines in the space). Although the approach is interesting and results seems good, the need to have a specific Laser mounting is a significant limitation to its use in real scenarios. In would be interesting to think about possible alternatives using more off the shelf HW or passive markers that would be much easier to have.

The mention of structure from motion algorithm in the related work is outside the scope, as author refers these methods are non-suitable for static camera setups (in which extrinsic calibration make sense). It would be interesting to discuss and compare the large physical object and the laser projected approaches. Authors refer that large physical object restricts the application to medium scale setups, but I guess laser projection also suffer of this limitation since in large environment it might be difficult to have several laser lines in all the images as the environment size scales with the cost of developing specific HW. It would be interesting to comment a little on that.

In the description of the algorithm, it would be interesting to use (or at least refer) the name of the steps. Just saying step B or C maker more difficult to relate to the algorithm step.

The relation between step B (line detection) and step C (plane optimization) is not obvious. How planes are estimated from the lines with two DoF? This could be further explained to help reader understand this important step. An additional figure (or adding the plane in figure 4) could probably support this explanation. This is important since in section 3.2 authors refer again lines with endpoints and not the planes estimation.  But then the plane estimation appears again in 3.3. I guess this is not difficult to understand but in the current description, it is not clear to follow.

In the monocular pose scaling, one of the possibilities proposed were to use to marker with known distance. Does it really require 2 markers with known distance? I guess markers with known size would do the trick as well without requiring evaluating a distance between the markers in the world.

As referred before the stereo calibration is an evolution of monocular, the block diagram of the processing for example is the same and all this section is quite redundant when compare with the monocular one.

At the beginning I was wondering about the use of synthetic data since at the end tests with real data are presented, but after reading the section and its evaluation of several parameters that can influence the calibration quality, I reconsider and believe this section is interesting.

One significant question of the method is the sensitivity to plane curvature: if no “good” planes are available this could significantly impact calibration quality of even the possibility to calibrate the system, right?

The selection of the other methods used in the real experiments’ validation (table 1) should be justified. Why these methods? Methods that allow calibration with no superposition of field of view? What was the main criteria for the presented selection? Also, the last bullet should be a new paragraph as it is not referring a calibration method.

In the conclusion I do not agree that the main advantage of the technique is that it does not require a checkerboard. The use of a checkerboard instead of a specific piece of HW for the projection of the laser would be beneficial IMHO. In my view its main advantage is the possibility to calibrate camera with non-overlapping views. This section should be reworded carefully to truly reflect the benefits of the method.

Round 2

Reviewer 1 Report

The presented paper is only slightly improved. I still can't see scientific novelty. All described algorythms are well known.